# Combining amino acid PET and MRI imaging increases accuracy to define malignant areas in adult glioma

Maciej Harat [1,2] ✉, Józefina Rakowska[3], Marek Harat[3,4], Tadeusz Szylberg[5], Jacek Furtak[3], Izabela Miechowicz [6] & Bogdan Małkowski [7,8] ✉

Accurate determination of the extent and grade of adult-type diffuse gliomas is critical to patient management. In clinical practice, contrast-enhancing areas of diffuse gliomas in magnetic resonance imaging (MRI) sequences are usually used to target biopsy, surgery, and radiation therapy, but there can be discrepancies between these areas and the actual tumor extent. Here we show that adding [18]F-fluoro-ethyl-tyrosine positron emission tomography (FET-PET) to MRI sequences accurately locates the most malignant areas of contrast-enhancing gliomas, potentially impacting subsequent management and outcomes. We present a prospective analysis of over 300 serial biopsy specimens from 23 patients with contrast-enhancing adult-type diffuse gliomas using a hybrid PET-MRI scanner to compare T2-weighted and contrast-enhancing MRI images with FET-PET. In all cases, we observe and confirm high FET uptake in early PET acquisitions (5–15 min after [18]F-FET administration) outside areas of contrast enhancement on MRI, indicative of high-grade glioma. In 30% cases, inclusion of FET-positive sites changes the biopsy result to a higher tumor grade.

Accurate determination of the extent and grade of gliomas is critical to patient management because grade, completeness of resection, and volume have prognostic and predictive significance. While conventional MRI is primarily used for planning glioma treatment, there may be value in using [18]F-fluoro-ethyl-tyrosine (18F-FET)-PET to improve the diagnosis and evaluation of gliomas[1,2].

Amino acid PET is particularly helpful for identifying malignant foci within non-contrast-enhancing gliomas for biopsy planning[1,3]. However, only a few small imaging studies have evaluated whether [18]F-FET uptake identifies higher grade tumors compared with grading using biopsies obtained by contrast MRI in patients with contrast-

enhancing gliomas[4–8]. Where there has been histopathological confirmation of imaging discrepancies, these have been determined used non-standard imaging algorithms[9] or they assessed heterogeneous patient populations[10], hampering clinical translation. Recently, hybrid PET-MRI scanners have been introduced that allow direct comparison of tumor extension in PET and MRI data, the main benefit being increased patient comfort[11–13].

It was shown almost twenty years ago that combined MRI and acquisition of [18]F-FET-PET images 20–40 min post radiotracer injection significantly improved diagnostic accuracy in patients with cerebral gliomas[10]. While tumor volumes determined using MRI or PET are

[1]Department of Neurooncology and Radiosurgery, Franciszek Lukaszczyk Oncology Center, Bydgoszcz, Poland. [2]Department of Oncology and Brachytherapy, Faculty of Medicine, Ludwik Rydygier Collegium Medicum, Nicolaus Copernicus University, Bydgoszcz, Poland. [3]Department of Neurosurgery, 10th Military Research Hospital, Bydgoszcz, Poland. [4]Centre of Medical Sciences, Bydgoszcz, University of Science and Technology, Bydgoszcz, Poland. [5]Department of Pathomorphology, 10th Military Research Hospital, Bydgoszcz, Poland. [6]Department of Computer Science and Statistics, University of Medical Sciences, Poznan, Poland. [7]Department of Nuclear Medicine, Franciszek Lukaszczyk Oncology Center, Bydgoszcz, Poland. [8]Department of Positron Emission Tomography and Molecular Imaging, Ludwik Rydygier Collegium Medicum, Nicolaus Copernicus University, Bydgoszcz, Poland. ✉e-mail: haratm@co.bydgoszcz.pl; malkowskib@co.bydgoszcz.pl

known to often significantly differ[4–6,8], there has yet to be a prospective, pathologically-verified analysis of combined $^{18}$F-FET-PET and MRI with contrast enhancement for biopsy and treatment planning[1].

The timing of PET image acquisition also matters. In our previous studies of glioblastoma, we showed that early (5–15 min after radiotracer injection) and standard timepoint PET acquisition images (20–40 min after radiotracer injection) differ in size and volume, allowing the identification of additional tumor recurrence sites after oncological treatment[14]. Furthermore, dynamic PET acquisition has been shown to be of value in improving diagnostic accuracy[15–18]. A failure to consider early acquisition images might limit or alter the results of glioma treatment.

Therefore, we conducted a prospective study of malignant glioma infiltration based on the histopathological analysis of serial stereotactic biopsies planned by PET-MRI, taking advantage of early PET acquisition 5–15 min after radiotracer injection. The main aim of the study was to compare the extent and grade of contrast-enhancing adult-type diffuse gliomas detected by MRI and FET-labeled PET through the histopathological assessment of prespecified targets. In doing so, this work shows that there is extension beyond MRI T1-Gad enhancing areas in 100% cases and, when there is PET extension beyond the MRI uptake, the histopathological diagnosis better represents the highest tumor grade. This improved biopsy accuracy in 30% of cases is likely to affect the clinical management of a significant proportion of patients.

## Results

The baseline characteristics of the study participants are shown in Table 1. In all cases, high FET uptake was visualized outside areas of contrast enhancement on MRI, and intraoperative histopathological examination confirmed glioma (Fig. 1a). Figure 1b shows representative examples of the four target areas: Target 1 (T1-GAD), taken from site of contrast enhancement in T1 MRI sequences and simultaneous hotspot FET uptake in PET images 5–15 min post-injection; Target 2 (PET⁻), site of contrast enhancement in T1 MRI sequences but outside PET hotspot; Target 3 (PET), hotspot FET uptake but without contrast enhancement on MRI; and Target 4 (FLAIR), peripheral areas hyperintense in T2 FLAIR without increased FET uptake or MRI contrast enhancement.

### Does early FET-PET reveal malignant tumor outside areas enhancing on MRI?

We first aimed to establish whether PET highlighted tumor areas outside areas of contrast enhancement. Overall, 306 samples were collected and 284 were examined: Target 1–95 samples (87 tumor, 4 astrogliosis, 4 excluded); Target 2–12 samples (8 tumor, 3 astrogliosis, 1 excluded); Target 3–116 samples (101 tumor, 9 astrogliosis, 6 excluded); and Target 4 – 83 samples (34 tumor, 38 astrogliosis,11 excluded) (Fig. 1a).

Only 41.5% of all tumor samples were found within contrast-enhancing areas (Targets 1 and 2). By comparison, 85% of all tumor samples were found inside areas of increased PET uptake (Targets 1–3). The remaining samples were found inside FLAIR-positive areas; however, the positive predictive value (PPV) of FLAIR was only 47.2% compared with 92% for Target 3 and 96% for Target 1. All sensitivity, specificity, PPV and NPV data are presented in Fig. 1a.

In all cases (100%), the most malignant tumor parts were pathologically confirmed outside contrast-enhancing areas in MRI, i.e., Target 3 (Supplementary Table S1). There was no correlation between final grade nor molecular characteristics and the presence of FET-positive sites outside contrast-enhancing sites. In two cases (8%), enhancement exceeded the hotspots in FET but did not affect the final histopathological diagnosis (Supplementary Table S1, Column C).

A quantitative uptake analysis of targeted points is presented in Fig. 1c. There was no difference between Target 1 vs. Target 3 but a significant difference between Target 1 and Target 3 vs Target 4 (Fig. 1c). Moreover, PET uptake outside FLAIR was higher than that inside FLAIR with pathologically-confirmed tumor; overall, hyperintense T2 FLAIR signal was smaller than the PET-avid area in 13 cases (56%), larger in six patients (26%), and comparable in four (18%) cases. A representative case of tumor uptake outside FLAIR is shown in Fig. 2a.

### Intratumoral heterogeneity

Tumors were heterogeneous in 74% of cases, with the diagnosis ranging from WHO II to IV depending on the site of biopsy in 60% of cases. A summary of the variability in individual tumors captured by the very rich dataset of over 300 biopsies is presented in Fig. 2b and Supplementary Table S1. There was no correlation between tumor heterogeneity and final tumor grade ($p = 0.5774$) nor the molecular characteristics.

### Tumor grade according to selected target

Overall, in seven out of 23 patients (30%), the biopsy result was changed to a higher tumor grade after the inclusion of FET-positive sites, including cases where material was obtainable from contrast-enhancing sites. The grade was unchanged in the remaining cases.

The concordances between each target and the final histopathological diagnosis are presented in Table 2. There was a high level of agreement between the final histopathological grade and the T1-GAD (Target 1) biopsy result ($\kappa_w = 0.69$; Table 2), and there was complete agreement between the final histopathological grade and the PET (Target 3) based biopsy result ($\kappa_w = 1.00$; Table 2). Overall, 110 samples were taken from Target 3, and, in 32 samples, a grade 4 glioma was

## Table 1 | Basic characteristics of the study participants

| Characteristic | | G2 (n = 2) | G3 (n = 10) | G4 (n = 11) | p-value two-sided |
|---|---|---|---|---|---|
| Age (years; mean, range) | | 38, 37–39 | 40, 18–48 | 56, 34–86 | 0.015[a] |
| Sex (n, %) | Female | 1, 8.3% | 6, 50.0% | 5, 41.7% | 0.828[b] |
| | Male | 1, 9.1% | 4, 36.4% | 6, 54.5% | |
| Previous treatment (n, %) | Previous radiotherapy | 0, 0.0% | 3, 50.0% | 3, 50.0% | <1.000[b] |
| | Untreated | 2, 11.7% | 7, 41.2% | 8, 47.1% | |
| Overall survival (median, IQR) | Diagnosis not changed | N/A | 52 (14-N/A) | 18 3–28) | |
| | Diagnosis upgraded by PET | N/A | 54 (54-54) | 7 (2-n/A) | |

Patient are divided according to tumor grade. Differences between each tumor grade in terms of age, sex, previous treatment, overall survival. Patients were similar in terms to basic characteristics according to each tumor grade with the exception of age (patients diagnosed with G4 were older.
N/A not applicable, SUV standardized uptake value; TBR, target-brain ratio p = 0.015)
[a]Kruskal–Wallis test with Dunn-Bonferroni multiple comparisons test.
[b]Fisher–Freeman–Halton test.

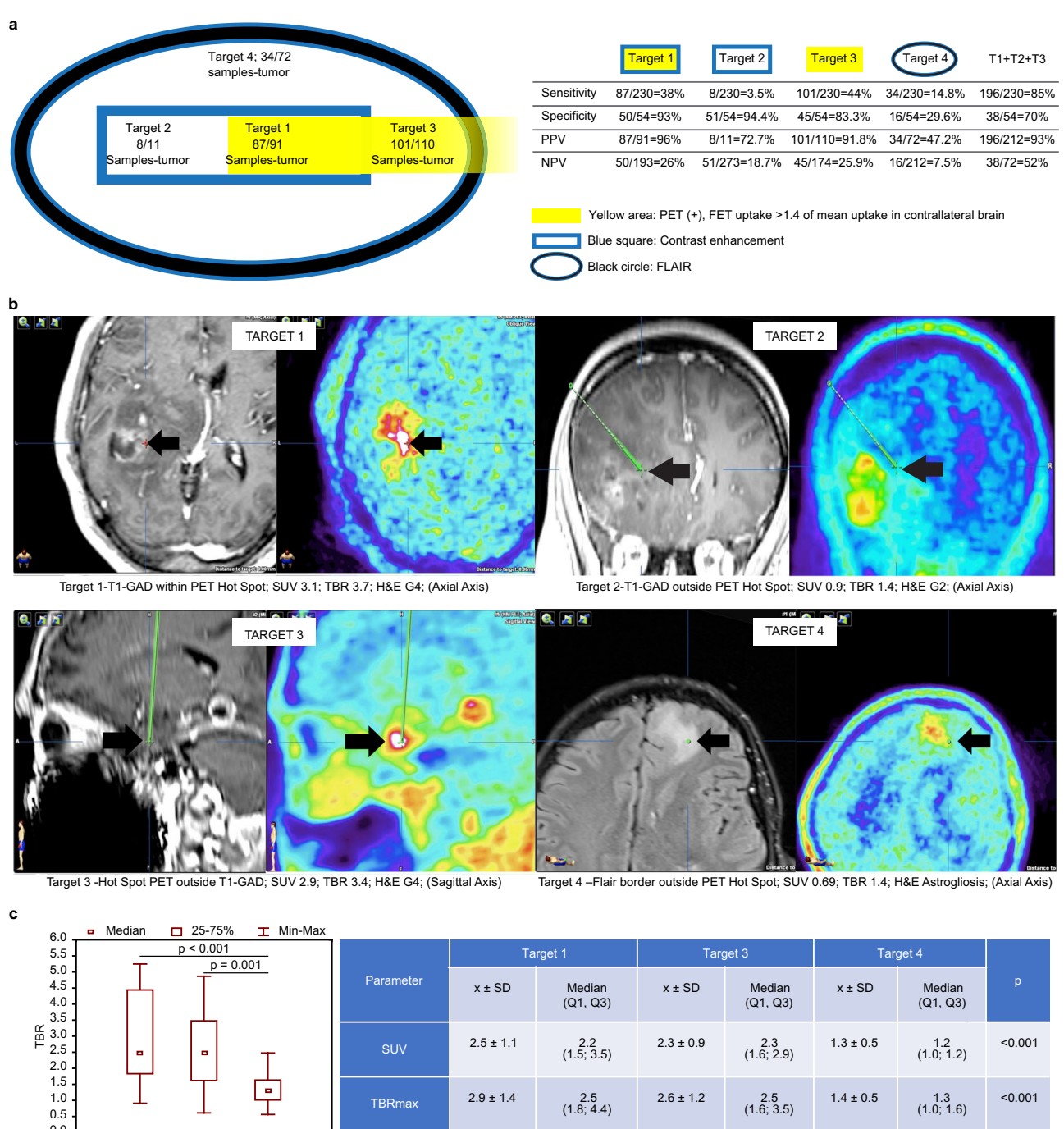

**Fig. 1 | Target definition and accuracy. a** Number of positive (tumor) samples of all samples analyzed from each target. The table shows the sensitivity, specificity, positive predictive values, and negative predictive values of all targets and the sum of Target 1, Target 2, and Target 3 to illustrate how tumor definition is improved by adding PET-positive areas. Yellow area represents FET uptake, blue square area represents contrast enhancement, and the black circle represents areas hyperintense in FLAIR. T1- target 1, T2 - target 2, and T3 - target 3. **b** Examples of targeted areas. Target 1 (T1-GAD): enhancement in T1-weighted MRI and uptake hotspot in FET-PET. Target 2 (PET-): contrast enhancement outside a PET hotspot. Target 3 (PET): hotspot FET uptake but without contrast enhancement in MRI. Target 4: border of a hyperintense area in FLAIR. Left columns show tumors on MRI, while the right columns show FET-labeled PET. Target 2 and Target 3 images show the

trajectory (green line) along which the material was collected. Black arrows and cross signs mark the biopsy targets. c Boxplot showing the comparison of tumor maximum uptake to brain (TBRmax) and standardized uptake values (SUV) in the targets (T1-Gad is related to Targets 1 and 2, PET is related to Target 3, and FLAIR to target 4). A comparison of uptake values revealed no difference in Target 1 (PET and T1-GAD) vs Target 3 (PET without GAD) but significant differences between Target 1 and Target 3 vs Target 4 (FLAIR). Differences between TBRmax values and SUV were determined using a two-tailed Friedman test with the Dunn-Bonferroni multiple comparisons test and is indicated with a line at the top of the graph. The number of patients analyzed is shown for each graph. Source data are provided in the Source Data file.

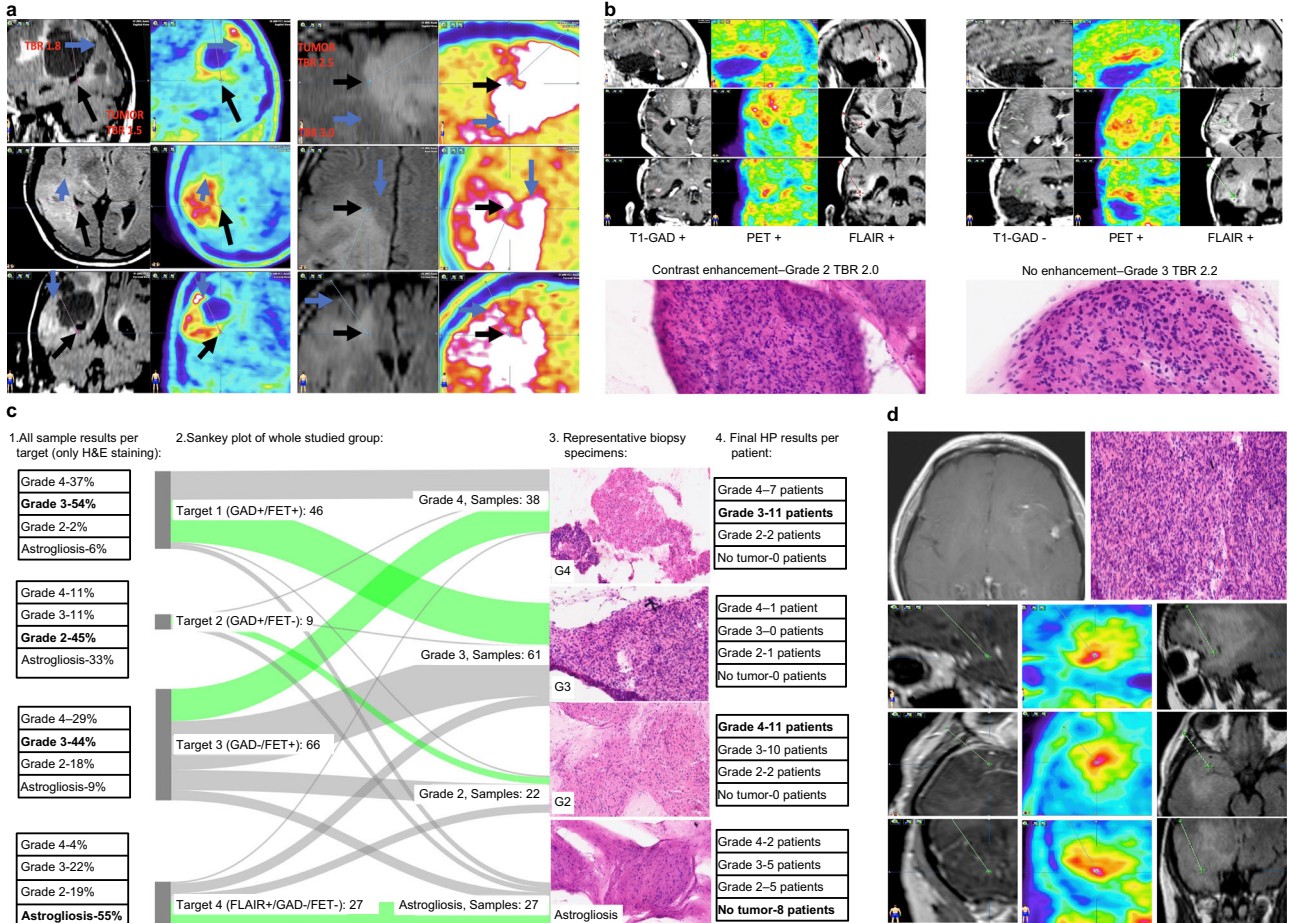

**Fig. 2 | Biopsy results from different target areas. a** FET-PET extension beyond hyperintense areas in FLAIR images correlated with TBR values. The black arrows represent tumor tissue in 3-plane images confirmed by the neuropathologist at the FLAIR border. Biopsy trajectories are marked by thin dotted lines on each image. The blue arrows show areas of higher TBR values extending beyond FLAIR hyperintense areas. In the entire cohort, in 56% of cases, stronger FET uptake was found outside FLAIR. **b** FET-uptake outside contrast enhancement and intratumoral heterogeneity. FET-PET helps to identify areas of highest grade outside contrast enhancement (*n* = 5). A representative case of upgrading from Grade 2 based on contrast-enhancing MRI (left side of the figure) to Grade 3 based on early FET uptake (right side). Photomicrographs of H&E-stained sections (200x magnification) showing increased cell density, nuclear atypia, and mitosis in Grade 3 relative to Grade 2 samples. **c** Sankey plot: 1. All samples evaluated by H&E (*n* = 148) per target (left column), Target 1 (upper row) to Target 4 (lower row). 2. Results from each target (in the middle). 3. Representative biopsy specimens. 4. All final

histopathology results per patient related to each target (right column), Target 1 (upper row) to Target 4 (lower row). The light green flow in the Sankey plot shows the most common diagnosis from targeted points per patient and highlights that Target 3 increases the Grade 4 final diagnosis and decreases the rate of astrogliosis obtained outside T1-GAD⁺ targets. Photomicrographs of the biopsy specimens (x100 magnification, H&E staining). Grade 4 (G4) specimen showing characteristic histopathological features of cellular density, necrosis, and polymorphous neuronal elements; G3 specimen showing increase cell density, mitoses, but no necrosis; G2 specimen with lower cell density, and astrogliosis found at the FLAIR border. **d** FET-PET in high-risk biopsy location, *n* = 3 (photomicrograph of H&E-stained section, 200x magnification). A contrast-enhancing target (Target 1) was not biopsied due to proximity to the branches of the left middle cerebral artery (left side), but Target 3 was a safe alternative (right side of the figure with trajectory visible as thin dotted line). Source data are provided in the Source Data file.

**Table 2 | Concordances between each biopsy site according to imaging modality and the final histopathological diagnosis**

| Target | n | weighted Kappa coefficient of concordance (linear weights) | 95%CI | Z test *p*-value two-sided | Correctly classified cases, *n* (%) | Incorrectly classified cases, *n* (%) |
|---|---|---|---|---|---|---|
| T1-GAD | 20 | 0.69 | 0.42; 0.96 | <0.001 | 16 (80%) | 4 (20%) |
| PET | 23 | 1 | | <0.001 | 23 (100%) | 0 (0%) |
| FLAIR | 20 | 0.29 | 0.03; 0.55 | 0.033 | 6 (30%) | 14 (70%) |

found (28%). Astrogliosis was only diagnosed in 9/110 samples (8%), but in all cases tumor tissue was also detected in the next sample taken. Final histopathological results in PET areas (Target 3) revealed that eleven cases were grade IV, 10 cases were grade III, and two cases were grade II glioma (Supplementary Table S1). A Sankey plot depicting the change in classification of each tumor based on the different regions biopsied is presented in Fig. 2c.

In three cases (13%), no material was collected from T1-GAD (Target 1) areas (Supplementary Table S1), including two cases due to proximity of vessels and the risk of serious complications and in the remaining case due to subtle and diffuse enhancement making it difficult to select the biopsy target. However, all patients showed increased uptake in early FET images, confirming the presence of neoplastic cells, and PET examination allowed the collection of

material from safer sites. There were no complications after biopsy. Figure 2d shows how FET provided targets outside contrast-enhancing lesions and decreased the risk of biopsy.

There was a low level of agreement between the final histopathological results and the results obtained from FLAIR hyperintense areas ($\kappa_w$ = 0.29; $p$ = 0.033). In three cases, it was not possible to define the tumor periphery based on FLAIR at all, hence a lack of collected material. In ten patients, reactive astrogliosis was detected, including three cases where astrogliosis and tumor tissue were both detected in various FLAIR samples. Across the entire study, 83 samples were taken from FLAIR areas, of which 11 were inconclusive and 34 contained tumor tissue (47%). All other samples collected from FLAIR regions were reactive astrogliosis. Overall, serial biopsies based on the FLAIR sequences provided a glioma diagnosis in 13 out of 20 examined cases (65%), most often of lower grade than in areas of high FET uptake. The histopathological result from the FLAIR was the same as the final histopathological result in only six cases (30%).

1p/19q codeletion and glioma cells present in the material collected from FLAIR-positive sites were correlated ($p$ = 0.04).

## Discussion

While other studies have confirmed the value of FET-PET-based biopsy in non-contrast enhancing gliomas[3,19], here we confirmed that FET-labeled PET identified the highest-grade glioma infiltration beyond contrast-enhanced MRI images in every case of adult-type diffuse glioma, with all results confirmed histopathologically. The low sensitivity of T1 contrast-enhancing MRI, low PPV of FLAIR, and high PPV of FET-PET outside T1 contrast-enhancing areas support the hypothesis that target definition can be improved by using FET-PET but not FLAIR. Future studies should examine low to moderate FET-PET uptake inside FLAIR areas, since this might further increase the sensitivity and PPV in terms of glioma definition in this area. The data strongly suggest that uptake hotspots in early PET acquisition images obtained by hybrid PET-MRI can be used to locate the most malignant areas with high accuracy. Moreover, FET-PET-based biopsy increased the diagnostic yield and grading accuracy over MRI-guided biopsy of contrast-enhanced adult-type diffuse gliomas. Brain areas without increased FET uptake or contrast enhancement but hyperintense in T2 FLAIR sequences were not specific for malignancy, and FET uptake extended beyond T2 FLAIR regions. Treatment or biopsy planning based solely on MRI images always missed areas of high amino-acid uptake visible in PET and representing tumor cells. Although further research is still needed on the optimal margin for FET-PET-based radiotherapy and the optimal tumor-normal tissue threshold in early FET-PET acquisition, our results support a new paradigm for planning treatment in patients with contrast-enhancing gliomas.

This study shows that FET not only defines infiltration beyond areas seen in MRI sequences but that the uptake intensity is similar to the uptake seen in T1-GAD areas, as shown in quantitative analyses. The uptake was so evident that neurosurgeons could identify and target the area subjectively, and these areas corresponded to areas suitable for making the exact histopathological diagnosis. Although background assessment is essential when evaluating TBR values and quantifying biological tumor volume, the TBR calculation was not crucial in terms of our findings. For individual patients for biopsy planning at individual timepoints, as the targeted points were usually localized in the same part of the brain and the same background value was used for all targets (Fig. 3a), differences between targets with respect to absolute SUV and TBR were similar (see Fig. 1c).

Hybrid PET-MRI has several advantages. First, integrated PET-MRI images are directly and accurately fused. Moreover, the use of gadolinium contrast does not interfere with PET acquisition and allows comparison of PET in relation to MRI without a time interval that may negatively impact the results. This is of particular importance in neuronavigation and biopsy planning. Second, the procedure favors patient comfort, which is particularly important in this population with decreased neurological function. The dual time-point [18]F-FET-PET protocol is of sufficient length to simultaneously acquire a multiparametric MRI of the brain without increasing the time required to conduct standard dual time-point PET or MRI. Finally, more accurate diagnosis and optimized treatment may help to reduce costs.

Differences in tumor extent identified by MRI or FET-PET have previously only been reported in limited retrospective or imaging studies[4–8], and these studies did not compare the imaging findings with gold standard histopathological assessment, nor were they prospective. Verburg et al. recently reported the improved detection of glioma infiltration using a combination of imaging modalities (apparent diffusion coefficient (ADC) mapping/FET-PET), but they only examined a small number of contrast-enhancing gliomas[9]. Our finding that early FET-PET combined with T1-GAD improves detection supports this hypothesis. Verburg et al. analyzed a series of 11 contrast-enhancing gliomas and most accurately detected glioma infiltration through a combination of ADC and [18]F-FET-PET (area under the curve (AUC) 0.95 vs. 0.76 for FET-PET alone), but the study was based on a standard acquisition protocol (20–40 min after FET injection) and did not include the early PET acquisition timepoint that impacts the detection of high-grade gliomas[20]. All biopsies in our study were planned using these early acquisition images taken 5–15 min after radionuclide injection. We previously showed that early acquisition images highlight larger areas of glioma infiltration than at standard acquisition timepoints[14]. Also, intrinsic FET kinetics may result in higher uptake at early timepoints[21], but this has not been confirmed histopathologically until now.

There remains some debate about the benefits of early FET and the risk of false positives, as early scans may be disproportionately impacted by the high FET concentrations present in the vascular pool just after injection[2]. Nevertheless, it is increasingly clear that tracer uptake varies over time depending on glioma grade. In a retrospective study correlating the results of 121 FET-PET examinations performed before histopathological assessment, uptake 12.5 min post-injection was significantly higher in grade 4 than grade 3 gliomas[22]. Hence, if the goal is to visualize highest-grade tumor sites, early PET examination should be used for biopsy. Supporting this, we previously showed that the histopathology obtained on the basis of early and standard PET images may differ[23], and the current data lend further weight to adopting the early timepoint PET protocol to improve the diagnostic yield.

In all cases, FET uptake outside contrast enhancement contained malignant glioma cells. Muther et al[24]. showed that, in 33 glioblastoma patients undergoing surgery, FET-positive areas were visible outside contrast-enhancing areas but also beyond areas defined by 5-aminolevulinic acid (5-ALA) used for fluorescence-guided resections. In another retrospective study, FET-avid regions after glioblastoma resection correlated with contrast enhancement occurring one to three months later[25]. Our pathologically confirmed results are also consisted with a previous image-based retrospective study showing that the tumor volume defined on contrast enhancement is smaller than the area defined by FET uptake in 86% of cases, while in five patients (10%) increased FET uptake was present outside areas of FLAIR hyperintensity[26]. Another retrospective study based on standard acquisition reported that the [18]F-FET uptake was located outside contrast-enhancing areas in 61% of cases and outside hyperintense areas in FLAIR images in 35% of cases[22].

The correct histopathological diagnosis is crucial for contrast-enhancing glioma management. Currently, the oncological management of grade 3 and 4 gliomas differs, for example with respect to qualification for management with tumor treating fields[27]. There have been previous histopathological analyses of biopsies from gliomas obtained on the basis of FET-PET. Pauleit et al. performed histopathological analysis of gliomas biopsied 20–40 min after injection by

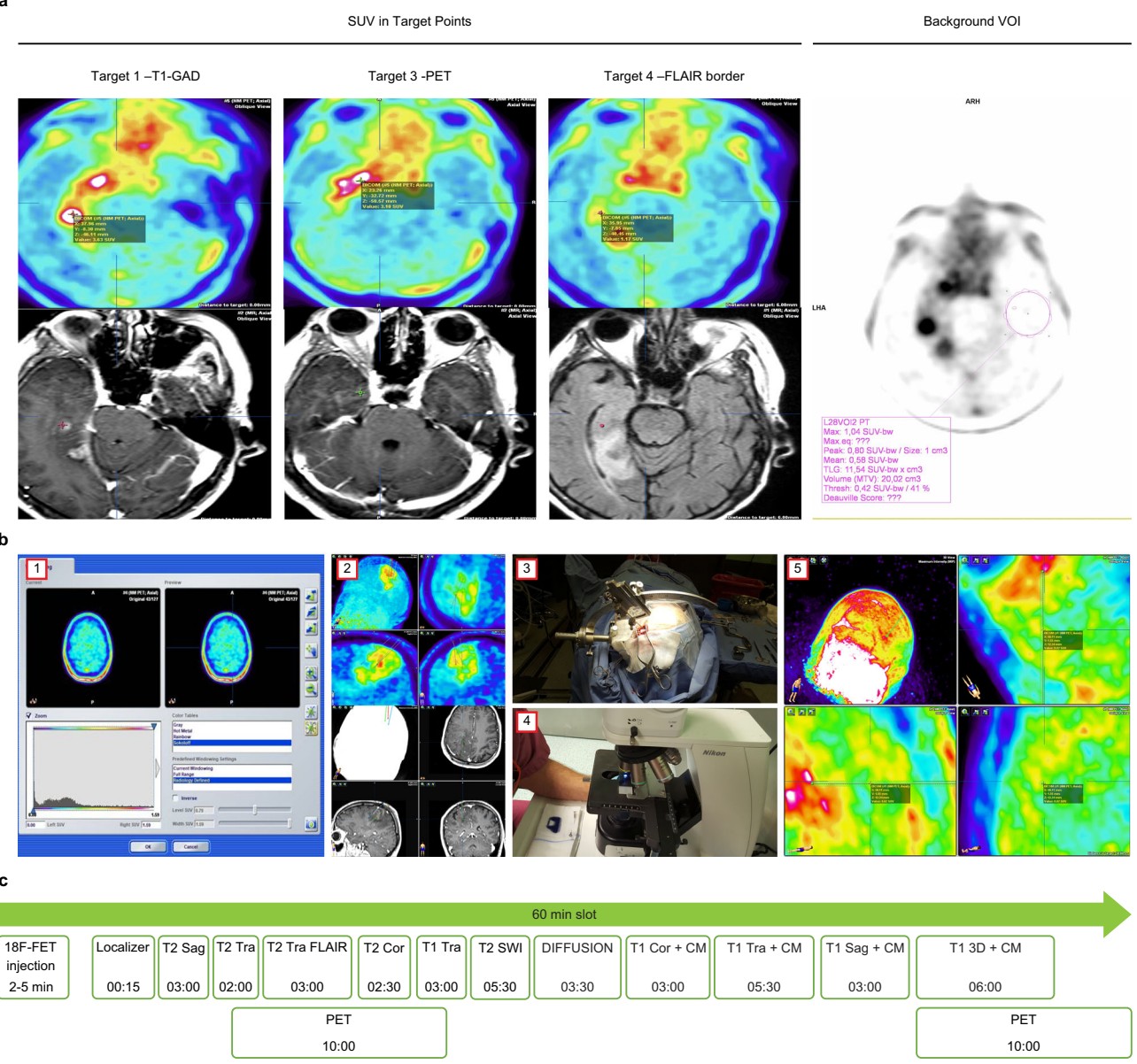

**Fig. 3 | Study methodology and the qualitative and quantitative analyses.**
**a** Quantitative SUV assessment in all targets and inside the contralateral brain. Background volume of interest (VOI) was used to quantitative analysis. **b** PET-MRI-guided stereotactic serial biopsy procedure. 1. Uptake values and fusion image analysis using iPlan (Brainlab) in the operating suite. 2. Biopsy trajectory planning.
3. Stereotactic serial biopsy. 4. Intraoperative histopathological analysis of selected samples. 5. Quantitative FET-PET image analysis at each target point. **c** PET-MRI protocol with dual point FET-PET acquisition. All abbreviations of MRI sequences are presented in Methods section.

FET-PET, but MRI and PET were performed separately and the study cohort was clinically heterogenous (included circumscribed gliomas and non-tumoral lesions)[10]. Only 10 of 26 biopsies containing tumor tissue contrast enhanced on MRI. While this study is cited in current recommendations advocating PET for defining glioma[16], our current analysis now provides further evidence by reporting glioma extent and heterogeneity in a clinically distinct population (adult-type diffuse gliomas) and suggests benefit from adopting the early acquisition timepoint more specific for high-grade gliomas using newer technology (hybrid PET/MRI).

A high level of evidence is needed to guide clinical practice when defining the tumor extent for radiotherapy or surgery[28], and although PET appears to provide the valuable data needed for surgery or radiotherapy, this recommendation is based on small datasets[9,10]. Our study now adds to this evidence base. PET/

Response Assessment in Neuro-Oncology (RANO) recommendations on the clinical use of PET in patients with gliomas highlight the value of amino-acid PET for defining glioma extent. However, these recommendations were based on a prospective pilot F-DOPA study on only ten patients[29], so the adoption of these recommendation has been limited. A current PET/RANO report on amino acid PET for radiotherapy of gliomas proposed that tumor delineation using amino acid PET might more accurately disclose the true tumor volume beyond that visualized by conventional MRI and identify additional tumor parts that should be targeted by irradiation (level of evidence 2)[3,7,9,10,30]. Indeed, the authors stated that "PET could hold the potential to detect tumor beyond what is achievable by conventional MRI"[16]. While there appears to be growing consensus that PET confers advantages for radiotherapy planning, the evidence has precluded its definitive recommendation for the

interdisciplinary planning of surgery or radiotherapy in patients with high-grade adult-type gliomas (e.g., in NCCN CNS 2022 guidelines[31]).

A recent study demonstrated improved survival in glioblastoma when the resection was extended beyond the area of enhancement into the T2/FLAIR abnormality, paving the way for not restricting surgical and radiotherapy target contours to only the contrast-enhancing portion identified on MRI[32]. Also, tumor resection based on FET-PET has been associated with a better prognosis in patients with higher-grade gliomas[33], and residual tumor volume in post-operative PET-FET images was an important prognostic factor in patients with glioblastoma[34–36]. Recently, we also showed encouraging results of radiotherapy with a boosted radiation dose based on early and standard acquisition[37]. In the current study, the hyperintense T2 FLAIR signal was poorly correlated with FET-PET images and in many cases (56%) it missed hot-spot uptake. We confirmed that glioma cells were found in hyperintense areas in FLAIR sequences in 65% of cases but in less than a half of samples collected from this area. These results show that the FLAIR sequence does not clearly identify the tumor border or glioma infiltration, as a variety of diagnoses were obtained from FLAIR (from the absence of neoplastic cells through to grade 4 glioma). The correlation between the presence of a codeletion and the presence of a neoplasm in FLAIR sequences is an unexpected result in a very limited number of patients, but it generates the testable hypothesis that FLAIR is more accurate in tumors with this molecular profile.

We recognize that our study could be limited by including patients after radiotherapy. This group of patients may show contrast enhancement as post-irradiation changes and, without pathological confirmation of tumor, these could represent false-positive results. However, all patients had histopathological confirmation of tumor and were therefore included in the analysis. The study was also performed before the new WHO 2021 classification of nervous system tumors was introduced, which mandates further genetic testing (including homozygous *CDKN2A/B* deletion for *IDH*-mutant astrocytomas and *TERT* promoter mutation, *EGFR* amplification, concomitant gain of chromosome 7 and loss of chromosome 10 (or just 10q) for *IDH*-wildtype astrocytomas). While this may impact the final diagnosis of nine of our cases, retrospective testing was unsuccessful. Regardless, the study provides important data on histopathological differences between MRI and PET regardless of the underlying molecular alterations. While determining biopsy areas visually may be regarded as a limitation, this approach facilitates the widespread implementation of the protocol in clinical practice for biopsy planning. Areas determined by visual analysis of high FET uptake are clearly an exact biopsy target, since a diagnosis was possible in every case that correlated with the final histopathological results. Preliminary quantitative image analysis found that a PET-MRI intensity threshold 1.4-times greater than symmetrical brain uptake could identify "FET hot-spot" areas, and this result now needs further validation. Although "standard" images (after 40–60 min) were acquired, these results are not presented, as early images were the novel aspect of the protocol, and the value of early acquisition remained a matter of debate when planning the study. Additionally, after biopsy, the group was found not to be homogenous (including few lower-grade gliomas) due to the intrinsic heterogeneity of contrast-enhancing gliomas. However, all patients qualified for the study prior to biopsy based on criteria suggesting high-grade glioma and all with contrast enhancement and other radiological characteristics of diffuse adult-type gliomas.

In conclusion, our data suggest a new approach for treatment planning of contrast-enhancing adult-type diffuse gliomas. Surgery and radiation therapy based on MRI alone is likely to exclude high-grade tumors from the treatment protocol. The data suggest that radiation therapy or surgery should be planned using early timepoint FET-PET. This was not, however, an interventional study; the research was not intended to impact the clinical decisions or further patient treatment. The aim of the study was solely to analyze differences between MRI and PET not connected with further interventions. Future studies will now need to focus on the irradiation margin needed when FET is included and exact target definition based on quantitative analyses of early and standard acquisition in high-grade diffuse gliomas.

## Methods

### Ethics approval
The institution's bioethics committee of Collegium Medicum Nicolaus Copernicus University approved the study (KB 647/2015), and informed consent was obtained from all participants.

### Study participants and inclusion and exclusion criteria
Twenty-three patients attending the Department of Neurosurgery, 10th Military Research Hospital, Bydgoszcz, Poland for stereotactic biopsy due to suspected adult-type diffuse glioma having undergone contrast-enhanced MRI were enrolled in this prospective study.

The inclusion criteria were: age >18 years; suspected CNS glioma based on MRI; and in general good health permissive of surgery. The exclusion criteria were: age <18 years; KPS < 60 points; pregnancy; and disqualification from surgery for medical reasons.

The reasons for performing CNS diagnostics in these patients included: headaches, epileptic seizures, cerebellar syndrome, speech disorders, and paresis. In a few patients, the tumor was detected accidentally due to diagnostics performed for a head injury. The basic characteristics of the participants are shown in Table 1.

### Pre-biopsy hybrid PET-MRI acquisition and evaluation
Patients meeting the inclusion criteria were referred for hybrid $^{18}$F-FET-PET-MRI examination in one procedure with simultaneous data collection at the Department of Nuclear Medicine, Franciszek Lukaszczyk Oncology Center, Bydgoszcz, Poland. PET was performed 5–15 min after administration of $^{18}$F-FET and again after about 60 min (dual time-point examination) as per standard procedures of the Department of Nuclear Medicine (Fig. 3b, c). For the $^{18}$F-FET-PET, all patients fasted for at least 4 h before PET-MRI as recommended, and a static emission recording in 3-dimensional mode was started on intravenous injection of 300 MBq of $^{18}$F-FET. The FET-PET acquisition time was 600 s, and the HD-PET (OSEM 3D-PSF, iterative-point spread function, 3 iterations, 21 subsets), relative scatter method, and filter Gaussian (FWHM 2.0 mm) algorithms were used.

Static PET data were reconstructed according to our clinical protocol using a 3-dimensional ordered-subset expectation-maximization algorithm with corrections for attenuation, scatter, random events, and dead time.

All $^{18}$F-FET-PET-MRI examinations were performed simultaneously on a 3-tesla PET/MR (Siemens Biograph mMR syngo MR E11; Siemens Healthineers) for the clinical indication. Patients underwent MRI scans with a slice thickness of 1 mm. In all patients, axial T2-weighted sequences as well as T1-weighted sequences before and after intravenous administration of a 0.1 mmol/kg bolus of gadobutrol (Gadovist [Bayer Healthcare]; injection rate of 3 mL/s, ACIST EMPOWER MR) were acquired. A dedicated ultrashort-echo-time sequence provided by the vendor was used for PET attenuation correction. The MRI acquisition was performed after early PET acquisition, and most sequences were completed before late PET acquisition at 40–60 min. A whole PET-MRI procedure showing all sequences is presented in Fig. 3b, c.

A standard MR imaging protocol comprising T2 TSE (T2WI, transverse, repetition time/echo time = 6000/96), T2 BLADE COR (T2WI, coronal, repetition time/echo time = 5500/118), T2 TIRM DARK FLUID (fluid-attenuated inversion recovery-FLAIR, repetition time/echo time = 6000/94), T2 SWI 3D (susceptibility-weighted imaging, 3-

dimensional, repetition time/echo time = 2200/2.5), DIFFUSION (diffusion-weighted imaging, B0, B1000, ADC, repetition time/echo time = 7100/95), T1 TSE TRA (T1WI, transverse, repetition time/echo time = 500/9), T1 TSE GAD (T1WI, transverse, sagittal, coronal, repetition time/echo time = 500/9) + CM, T1 MPRAGE 3D GAD (T1WI (magnetization prepared rapid gradient echo), 3-dimensional, repetition time/echo time =;2200/2,48) + CM was used.

[18]F-FET-PET image analysis included the evaluation of both static images. The PET-MR analysis conducted prior to biopsy was fully quantitative using dedicated software (Syngovia VB60, Siemens Healthcare, Erlangen, Germany). Based on VOI analysis, mean uptake values in the symmetrical brain and SUV values in voxels corresponding to target points were defined. A spherical VOI (diameter 30 mm) in the contralateral hemisphere including white and gray matter was used[20,38]; however, the SUV in the final target was defined as a point value (see Fig. 3a).

## Selection of biopsy site in the operating theater

The final biopsy site was selected by two experienced neurosurgeons. A first visual analysis was a qualitative evaluation, where the lesion of interest was classified as either positive or negative, the former applying when tracer uptake visually exceeded the background activity in the contralateral cortex. Biopsy target points were selected by analyzing the uptake value in relation to contrast enhancement and the uptake in surrounding areas. FET uptake was visualized at the time of biopsy target determination as representing the same or higher tracer uptake (a hotspot, Target 3) to uptake within contrast enhancement (Target 1). Our previous study[14] revealed major differences in T1-Gad and early FET-PET volumes, so we prespecified this target area prior to starting this study.

We did not select an exact threshold since this value was unknown for early acquisition. If the hotspot (similar or higher uptake than uptake within contrast enhancement) was found outside the contrast enhancement, the area was selected. Therefore, the PET-MR threshold method for determining target points in the operating theater was qualitative, but a quantitative analysis with TBR values (mostly >1.5, median 2.5) was also performed. An exact SUV in biopsy target points was defined in the operating theater at each target site using the same iPlan software as for biopsy. Quantitative PET image analysis was performed retrospectively based on iPLAN results and compared with prospectively assessed values in the Nuclear Medicine Department using dedicated software (SyngoVia). Maximum SUV (SUVmax) was measured in PET images. SUV values imported to iPlan were the same as those obtained directly from the PET scanner (Biograph MR, Siemens). Finally, the SUV in the target was defined as a point value (as presented in Fig. 3a; 3b5). Target-to-brain values (TBR = target/brain) were defined as a ratio between values measured at each target point and a mean uptake defined from the contralateral brain. TBRs and SUVs were compared in the whole study group with a post-hoc correction for repeated measures.

## PET-guided stereotactic biopsy procedure

The median time from PET to biopsy was seven days (IQR 3–13 days). Before biopsy, each patient had: [18]F-FET PET-MRI; routine MRI of the head for neuronavigation (T1 and T1 sequences with gadolinium contrast, T2 and T2 FLAIR sequences); routine chest X-ray for surgery; electrocardiography; routine pre-operative blood tests; and head CT (stereo thin layers) with the stereotactic frame attached. The stay at the Department of Neurosurgery lasted four days.

A routine biopsy procedure was used except for selecting targets based on fused PET-MRI images rather than MRI alone, with more samples taken than usual for histopathological examination. In some cases, it was necessary to plan several trajectories with an extra trepanation hole. The procedure was performed under local anesthesia in analgosedation.

The frame was first fixed to the patient's head prior to stereo CT examination and patient transfer to the operating theater. Using BrainLab software (iPlan Stereotaxy 3.0, Munich, Germany) in the operating theater, CT, MRI, and PET images were fused and the biopsy target parameters determined. Biopsy sites were selected as described above. The following biopsy sites were determined in each case: Target 1 (T1-GAD), taken from site of contrast enhancement in T1 MRI sequences and simultaneous hotspot FET uptake in PET images 5–15 min post-injection; Target 2 (PET⁻), site of contrast enhancement in T1 MRI sequences but outside PET hotspot; Target 3 (PET), hotspot FET uptake but without contrast enhancement on MRI; and Target 4 (FLAIR), peripheral areas hyperintense in T2 FLAIR without increased FET uptake or MRI contrast enhancement (Fig. 1b).

It was not always necessary to plan four trajectories to collect the material, since it was sometimes possible to collect different samples on one trajectory. For example, if on a given trajectory the area of isolated FET uptake was more superficial and the MRI contrast enhancement was deeper, material corresponding to two different targets could be taken from one trajectory but at different depths. An example of biopsy trajectories is presented in Fig. 1b.

Each target was defined as a 0 point measuring a few millimeters in area, i.e., the target ranged, for example, from −6 to 0, where material was taken every 1 mm from −6 depth to 0 along a planned trajectory. Every first sample was tested intraoperatively, and every second sample was taken for histopathological and molecular examination. Approximately six serial biopsies were taken in areas of contrast enhancement, increased FET uptake, and peripheral areas. Therefore, in total, about 18 samples were obtained from each patient, i.e., ~400 samples in total for the entire cohort. The highest-grade malignancy detected at each serial biopsy site is presented in the results.

After planning, the patient's head was attached to the operating table through the previously installed frame. To avoid brain shift during biopsy, the patient was positioned to achieve a burr hole location in the most supine point of the cranium. The x, y, and z coordinates of the entry point were calculated by the stereotaxis program, thus determining where the head should be shaved, the skin incision, and the site for the trepanation hole. After washing and preparation, the site was anesthetized with 1% lignocaine solution and a small skin incision was then made to expose the skull bones. With a manual trepan, and for tumors in the posterior cranial fossa, a small trepanation was made with a high-speed drill. Then, the visualized dura mater was coagulated, incised, and, in accordance with the calculated parameters, a cannulated biopsy needle was placed in the brain. Forceps were inserted through the cannula and several small samples taken for histopathological examination at 1 mm intervals. A neuropathologist was present in the operating room throughout the procedure, who initially assessed the material. The remaining tumor fragments were placed into sample tubes with formalin and then processed and assessed by a pathologist.

## Histopathological and molecular evaluation

A histopathological diagnosis was made intraoperatively with complete histopathological assessment performed postoperatively. Gliomas were graded according to the 2021 WHO classification.

For intraoperative evaluation, tissues were smeared and crushed onto glass slides before staining with methylene blue and microscopic examination. Postoperatively, formalin-fixed, paraffin-embedded tissue samples were stained with hematoxylin and eosin for microscopic examination, with immunocytochemical and immunohistochemical methods used for diagnosis where applicable.

For molecular evaluation, biopsy samples were dewaxed. DNA was extracted using the magnetic method (Maxwell®16 FFPE Tissue LEV DNA Purification Kit, Promega, Madison, WI). The Plexor®HY System (Promega) was used to determine the DNA concentration. Molecular

testing became standard practice during the course of this study, so only some patients had genetic test results. The molecular methods used in our institution are described in ref. 39.

The same experienced neuropathologist (TS) performed all histopathological diagnoses blinded to the patient details. Tissue obtained by stereotactic biopsy was oligobiopsy material and therefore of small size; most samples did not exceed 1 mm in diameter. Features of malignancy included increased cell density, anaplasia, and degenerative changes such as necrosis (see Fig. 3). Due to the small amount of obtained tissue, the representative photomicrographs do not always contain the full set of histological features characteristic for a given type of tumor.

## Statistical analysis and reproducibility

All analyses were conducted in PQStat v.1.8 (PQStat Software, Poznan, Poland). A *p*-value of <0.05 was considered statistically significant. Fisher's exact test or the Fisher-Freeman-Halton test were calculated to investigate relationships between categorical variables. Cohen's kappa concordance coefficient (with linear weights) was calculated to determine the agreement of both assessments.

The normality of the distributions was tested with the Shapiro-Wilk test. To compare the variables, single-factor repeated-measures analysis of variance (ANOVA) with Tukey's multiple comparison test or the Friedman test with the Dunn-Bonferroni multiple comparison test were computed. The unpaired *t*-test or Mann–Whitney test were used to compare variables between two groups, while the Kruskal–Wallis test with the Dunn-Bonferroni multiple comparison test was calculated to compare variables between more than two groups. ROC analysis was performed to determine the TBR threshold (cut-off) for discrimination of normal and tumor tissue with DeLong's method. Sensitivity and specificity were determined using the Youden index.

Replication of the single tissue sample analysis was not performed however efforts to verify reproducibility were attempted by serial biopsy assessment and overall patients number. Multiple tumor samples per patient per target were evaluated and all results are presented in source data/ All techniques and reagents used for the analyses of this study had been previously optimized and validated in clinic. In general, please note that technical replicates from tumor biopsies were not performed due to limited sample volumes. Presented photomicrographs were double checked by blinded pathologist.

No statistical method was used to predetermine sample size. No data were excluded from the analyses. The experiments were not randomized. The investigators were not blinded to allocation during experiments and outcome assessment.

Findings apply to both sexes (11 male patients, 12 female patients), and sex and gender were not considered in the study design Sex was determined on self-reporting and collected in the source data. Consent was obtained for sharing of individual-level data. No compensation was given to the research participants that acknowledged the time and effort they provided in participating in the research.

## Reporting summary

Further information on research design is available in the Nature Portfolio Reporting Summary linked to this article.

# Data availability

All data generated in this study are presented in the paper, and the raw data are provided in the Supplementary Information file and the Source Data file. Source data are provided with this paper.

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

## Acknowledgements
The authors would like to thank all the study participants and their families as well as all nurses, physicians and other medical staff who were involved in the patients care. This work was supported by Nicolaus Copernicus University. No external funding was received for this study. Production are publishing costs were covered by all authors.

## Author contributions
Mc.H and M.H. designed the study; J.R., J.F., T.S. and B.M. performed the majority of the experiments; Mc.H. and I.M. contributed to the analysis and interpretation of results; Mc.H. wrote the first draft of the manuscript; Mc.H. and M.H. has approved the final manuscript and completed manuscript; and all authors agreed the final content of the manuscript.

## Competing interests
The authors declare no competing interests.
