## [Peer Review File · Nature Communications]

Combining amino acid PET and MRI imaging increases accuracy to define malignant areas in adult gliomaREVIEWERS' COMMENTS:

Reviewer #1 (Remarks to the Author): with expertise in glioma, imaging

In this manuscript, the authors reported that hybrid 18F-FET-PET-MRI can be used to locate the most malignant areas of gliomas with pathologic evidence. Intra-tumoral heterogeneity is a challenge in accurate biopsy target, especially in non-contrast-enhancing gliomas. This article aimed to determine the border and grade of malignant gliomas derived from FET-PET-guided biopsy samples with 1 mm intervals, with the comparison between FLAIR/T1Gad MRI sequences and 18F-FET-PET. I have the following comments:

1. It is known that malignant gliomas may infiltrate into a large area of brain parenchyma and the border of gliomas may not exist. Please better describe the purpose of the study.
2. The histopathologic exam was used as gold standard, so at least H&E-stained pictures captured under microscopy are needed paralleled to the PET-MRI images in three figures. In the methods section, please also add details about the histopathological diagnosis analysis criteria. In addition, 2021 WHO classification (not 2016) should be used.
3. FLAIR images are needed in three figures. It is quite to find some tumor cells outside of FLAIR-hyper-intensive areas. However, not sure whether it is a good idea to use this to define the border of the tumor. Please do this with the standard histopathological diagnosis analysis criteria.
4. Quantitative image analysis should be performed. Without a proper PET-MRI intensity threshold, it was totally unclear how to identify areas of "FET positive" or "FET negative". Indeed, based on Figure 1, these areas of so-called "FET positive" or "FET negative" were somewhat subjective.
5. The statistical analysis methods used were unclear, lacking details. A method to determine a proper PET-MRI threshold may be needed. Another issue is multiple samples were obtained from a same subject. Did the author perform post-hoc analysis correction for repeated measures? The statistical results listed in Table 1 apparently ignored subgroup evaluations and P values. Table 2 needs more editing on an appropriate layout.
6. The interval between the biopsy targets is 1 mm. What are the spatial resolutions for the images obtained from hybrid 18F-FET-PET-MRI? Could the PET-MRI precisely identify biopsy targets with 1 mm interval?
7. The brain shift is a typical issue during the neuro-surgery procedure. How did the authors correct for this error?
8. If the purpose of the study is to determine "the border and grade of malignant gliomas", gliomas treated with radiotherapy should NOT be included. These treated gliomas may have contrast-enhancing lesions that are due to treatment effect, instead of active gliomas. If these are included, the study would not be meaningful.

Reviewer #2 (Remarks to the Author): with expertise in glioma, clinical

In this study, the authors report on prospective study assessing the utility of using Amino acid PET-MRI imaging to improve the accuracy of needle biopsy from 23 patients with contrast-enhancing high-grade gliomas. Several studies have previously demonstrated that hybrid PET/MRI is superior to MRI alone in detecting malignant transformation of low grade tumors and differentiating low-grade vs. high grade tumors (Systematic review - Clinical and Translational Imaging, 2021;9;609-623). Harat et al. Provide only a small incremental increase in the literature with examination of patients specifically with contrasts enhancing regions.

Major points:

- 1) In this study, the authors identified eight patients whose biopsy result was changed to a higher grade. To strengthen the study, the authors should provide the specific grading of the tumor based on i) site of contrast enhancement + increased FET uptake, ii) site of contrast enhancement + no FET uptake, iii) site of increased FET uptake + no contrast enhancement and iv) T2 flair + no FET uptake or MRI contrast. A Sankey plot depicting the change of classification of each sample based on the different regions of biopsy would improve the article.
- 2) The authors should state whether the pathologist were blinded to the origin of the tissue

specimen (i.e. based on region with increased FET uptake). In addition, we're the biopsy samples collected in the same order (i.e. first from site of contrast enhancement and FET uptake, followed by site of contrast enhancement with no FET uptake, etc ..)

3) The authors should include the clinical outcomes of the patients and demonstrate that the 8 patients with tumor grade changed based on FET uptake regions behaved like higher grade tumors.

RESPONSE TO REVIEWERS' COMMENTS

Reviewer #1, with expertise in glioma, imaging

In this manuscript, the authors reported that hybrid ^{18}F -FET-PET-MRI can be used to locate the most malignant areas of gliomas with pathologic evidence. Intra-tumoral heterogeneity is a challenge in accurate biopsy target, especially in non-contrast-enhancing gliomas. This article aimed to determine the border and grade of malignant gliomas derived from FET-PET-guided biopsy samples with 1 mm intervals, with the comparison between FLAIR/T1Gad MRI sequences and ^{18}F -FET-PET. I have the following comments:

1. It is known that malignant gliomas may infiltrate into a large area of brain parenchyma and the border of gliomas may not exist. Please better describe the purpose of the study.

Thank you for this comment. We agree that the word “limits” used in our aim is suboptimal when applied to the infiltration of malignant gliomas, and of course the exact determination of glioma infiltration is challenging or even impossible based on current imaging technologies.

Therefore, we emphasize that the aim of the study was to correlate FET-PET and MRI features derived from a hybrid scanner with pathological evidence and to verify that the tumor extension visible on ^{18}F -FET-PET but not in T1-Gad MRI sequences is true tumor extension rather than a false positive. Secondary aims were to analyze by histopathology the hyperintense FLAIR border and uptake in this area and compare FLAIR images with PET in relation to uptake.

Better defining the tumor for surgery or radiation therapy should translate into improved treatment outcomes, although we accept that even then the treated field may be still far away from true tumor border, which may not even exist. Nevertheless, our main objective was to test standard imaging used for surgery and radiation therapy against an emerging technique not currently recommended to strengthen (or otherwise) the evidence base for using that technique.

2. The histopathologic exam was used as gold standard, so at least H&E-stained pictures captured under microscopy are needed paralleled to the PET-MRI images in three figures. In the methods section, please also add details about the histopathological diagnosis analysis criteria. In addition, 2021 WHO classification (not 2016) should be used.

We have modified Figures 1-3, which now include representative photomicrographs of the histopathology.

We have also now provided details about the histopathological diagnostic criteria on Lines 367-373.

However, as the study was performed before new WHO 2021 classification was introduced, we have clarified in the limitations that, in the absence of genetic testing (which we tried but was unsuccessful) according to the 2021 classification, the final diagnosis of nine cases could be altered. Nevertheless, the study provides important data on the histopathological differences between MRI and PET, regardless of these molecular alterations.

3. FLAIR images are needed in three figures. It is quite to find some tumor cells outside of FLAIR-hyper-intensive areas. However, not sure whether it is a good idea to use this to define the border of the tumor. Please do this with the standard histopathological diagnosis analysis criteria.

We have now added FLAIR images to all figures. We have also clarified that we did not target areas outside the FLAIR hyperintense areas but at the FLAIR border, as defined by two experienced neurosurgeons (please see new Supplementary Figure 3).

Based on uptake analysis and image comparison we found strong FET uptake (higher than at the FLAIR border with confirmed tumor) outside FLAIR hyperintense areas was present in 56% of cases (Supplementary Figure 2 presents an example). However, we did not target those areas, as we did not expect to detect this when we planned the study. The biopsies corresponding to FLAIR images contained normal as well as tumor tissue. Our conclusion is that our study provides crucial evidence that radiotherapy or surgery targets should not be based on FLAIR sequences due to low specificity, (i.e., glioma cells found in hyperintense areas in FLAIR sequences in around 50% of samples).

As an additional finding, the correlation between the presence of a codeletion and the presence of a neoplasm in FLAIR sequences is an unexpected result in a very limited number of patients, but it generates the testable hypothesis that FLAIR is more accurate for tumors with this molecular profile.

4. Quantitative image analysis should be performed. Without a proper PET-MRI intensity threshold, it was totally unclear how to identify areas of “FET positive” or “FET negative”. Indeed, based on Figure 1, these areas of so-called “FET positive” or “FET negative” were somewhat subjective.

Thank you for this comment. We have now clarified this in the current version of the manuscript. We agree that the quantitative analysis is extremely important and can provide readers with an objective criterion and the possibility of authentic replication of our results in practice.

In reality, we emphasize that all targets were selected based on qualitative analysis of the images without quantitative data, and we have elaborated on this approach in the revised manuscript.

However, we have also now performed a quantitative analysis. The TBRs were similar in T1-Gad and PET (PET positive – median 2.5) sequences and significantly lower at the FLAIR border (PET negative – 1.3, see new data in **Table 1**).

5. The statistical analysis methods used were unclear, lacking details. A method to determine a proper PET-MRI threshold may be needed. Another issue is multiple samples were obtained from a same subject. Did the author perform post-hoc analysis correction for repeated measures? The statistical results listed in Table 1 apparently ignored subgroup evaluations and P values. Table 2 needs more editing on an appropriate layout.

We have now added more details about our statistical approach – which did indeed include post hoc tests for repeated measures, on Lines 380-388.

We have also added exact SUVs and TBR ratios for each targeted area in Table 1 and Supplementary Figure 2. We have also edited Table 2.

6. The interval between the biopsy targets is 1 mm. What are the spatial resolutions for the images obtained from hybrid 18F-FET-PET-MRI? Could the PET-MRI precisely identify biopsy targets with 1 mm interval?

The biopsy interval was 1 mm and the MRI resolution in the PET/MRI hybrid scanner was also 1 mm, although the PET resolution was 4.2 mm. The software algorithms superimposed the SUV on the MRI image and we found it to be as precise as biopsy-confirmed results. Nevertheless, we agree that the exact SUV represents an approximation.

7. The brain shift is a typical issue during the neuro-surgery procedure. How did the authors correct for this error?

We made every attempt to avoid brain shift during biopsy by positioning the patient to locate the burr hole in the most supine point of the cranium (an example is shown in **Supplementary Figure 1**), thereby avoiding CSF leakage and air absorption into the cranial cavity, the main cause of brain shift during stereotactic biopsy. We also tried to incise the dura without leakage and then coagulate the arachnoid to leave it arachnoid intact. The biopsy trajectory was planned through the gyri and avoiding the sulci, fissures, and ventricular system. We follow the main concepts presented previously, for example in:

Coenen, V A et al. "Minimizing brain shift during functional neurosurgical procedures - a simple burr hole technique that can decrease CSF loss and intracranial air." *Central European neurosurgery* vol. 72,4 (2011): 181-5. doi:10.1055/s-0031-1279748

We have clarified these details in the revised manuscript on Lines 339-340.

8. If the purpose of the study is to determine "the border and grade of malignant gliomas", gliomas treated with radiotherapy should NOT be included. These treated gliomas may have contrast-enhancing lesions that are due to treatment effect, instead of active gliomas. If these are included, the study would not be meaningful.

We agree, but the main purpose of the study was not to show the true infiltration or grade of gliomas rather to compare MRI and amino-acid PET images with pathological confirmation to verify the value of the various modalities for biopsy, surgery, and radiation therapy planning clinical scenarios (newly diagnosed and recurrent tumors). We have now clarified this on Lines 239-244.

We only analyzed patients after irradiation if T1-Gad confirmed a glioma and excluded post-irradiation changes. We do of course understand that false positive tumor may be present in images after irradiation, but in our cohort of irradiated patients we were able to verify and prove true tumor recurrence in T1-Gad as well as PET images.

Therefore, we believe that our findings are reliable and robust as there was high PET uptake beyond T1-Gad areas with histopathological confirmation regardless of previous irradiation.

Reviewer #2, with expertise in glioma, clinical

In this study, the authors report on prospective study assessing the utility of using Amino acid PET-MRI imaging to improve the accuracy of needle biopsy from 23 patients with contrast-enhancing high-grade gliomas. Several studies have previously demonstrated that hybrid PET/MRI is superior to MRI alone in detecting malignant transformation of low grade tumors and differentiating low-grade vs. high grade tumors (Systematic review - *Clinical and Translational Imaging*, 2021;9;609-623). Harat et al. Provide only a small incremental increase in the literature with examination of patients specifically with contrasts enhancing regions.

We agree that there have been some previous studies in this domain, but we respectfully disagree that the study is only incremental in value. We are sorry that we failed to develop the narrative in such a way to systematically (but driven by the evidence) explain why our results are highly significant, both in their own right and with regards to clinical impact.

To our knowledge, ours is the largest, prospective study comparing MRI with amino acid PET of contrast-enhancing adult-type diffuse gliomas with histopathological confirmation of the diagnosis.

We make the highly significant finding that early acquisition PET reveals hot-spot uptake (median TBR values 2.5) representing tumor of highest grade outside MR contrast-enhancing areas in 100% of cases. This is the first report showing the value of early PET acquisition in comparison with MRI. Furthermore, this is the first paper to show that FET not only defines infiltration beyond MRI but that the level of uptake is similar to uptake in T1-Gad areas (as presented in the new quantitative analysis). The uptake was so evident that even subjective selection by the neurosurgeon identified and targeted the relevant sites and correlated with exact diagnosis. This is of significant practical clinical value.

The difference in tumor extent identified by MRI and FET-PET has been presented in limited retrospective studies, but until now there has not been histopathological confirmation in adult-type diffuse gliomas in such a comprehensive manner. This therefore speaks to a real clinical need to properly define the tumor extent for radiotherapy or surgery.

Where histopathological verification of tumor infiltration beyond T1-Gad and FLAIR areas has been provided in previous studies, this has only been of a very limited number of contrast-enhancing gliomas (n=11) using ADC/FET-PET, which is both weak in terms of the evidence and clinical applicability. The other study with pathological confirmation of the diagnosis of gliomas seen on FET-PET imaging was a highly heterogeneous group (including circumscribed gliomas or non-tumoral lesions) using later timepoint protocols. Our study now refines the protocol for a specific patient population to form the basis of for updating clinical guidelines, not least confirming the value of early timepoint PET acquisition. Given that previous guidelines for PET use in patients with gliomas has been based on very weak evidence (e.g., PET-RANO are based on a 10-patient F-DOPA prospective pilot study), our study has the very real possibility of informing clinical practice.

There is also growing clinical evidence that extending the surgical or radiotherapy field beyond the T2/FLAIR enhancement is beneficial in terms of survival, and our study now lends an evidence base to the pathology driving this clinical observation.

Finally, FET also has an impact on biopsy results of contrast-enhancing diffuse gliomas. PET has generally not been recommended for biopsy of contrast-enhancing gliomas due to a lack of evidence that PET changes the final diagnosis. Our study now provides prospective, well-defined data showing that FET-PET discriminates grade 3 from grade 4 gliomas, and this difference will clearly have an impact on neuro-oncological practice.

We have re-written the Abstract, Introduction, and Discussion to better present these critical points.

Major points:

1) In this study, the authors identified eight patients whose biopsy result was changed to a higher grade. To strengthen the study, the authors should provide the specific grading of the tumor based on i) site of contrast enhancement + increased FET uptake, ii) site of contrast enhancement + no FET uptake, iii) site of increased FET uptake + no contrast enhancement and iv) T2 flair + no FET uptake or MRI contrast. A Sankey plot depicting the change of classification of each sample based on the different regions of biopsy would improve the article.

We have now provided a Sankey plot in the revised manuscript presenting all histopathological results obtained from targeted points (**Figure 4**).

2) The authors should state whether the pathologist were blinded to the origin of the tissue specimen (i.e. based on region with increased FET uptake). In addition, we're the biopsy samples collected in the same order (i.e. first from site of contrast enhancement and FET uptake, followed by site of contrast enhancement with no FET uptake, etc ..)

The histopathologist was blinded to the PET results and did not know from what site the specimen was taken. Furthermore, the biopsy target order was random and mainly was related to the trajectory of the biopsy needle. We usually used two or three trajectories. We have now added these details to the paper and clarified the methods.

3) The authors should include the clinical outcomes of the patients and demonstrate that the 8 patients with tumor grade changed based on FET uptake regions behaved like higher grade tumors.

We now detail the overall survival of all cases in **Table 1**, which shows that survival was longer in *IDH*-mutant tumors and some of the cases re-staged as grade 4 had very limited survival.

REVIEWER COMMENTS

Reviewer #2 (Remarks to the Author):

Reviewers have addressed all comments,

Reviewer #3 (Remarks to the Author):

NCOMMS-22-22647A-Z

Amino acid PET-MRI of contrast-enhancing adult-type diffuse gliomas

A reviewed manuscript was available for evaluation which investigated the value of FET-PET-MR in 23 patients with a heterogenous group of patients with brain tumors. Here, early FET images were compared with corresponding contrast-enhanced and T2-weighted images and correlated with tissue sampling from stereotactic biopsy.

Overall, this is an interesting study that provides some additional evidence to what is already known from the available literature. Even though the manuscript seems interesting in principle, in my opinion major revisions seem necessary:

First, I must point out that in the revised manuscript, the changes have unfortunately not been marked, as is normally the case. In this respect, I can unfortunately also not understand the corrections mentioned in the rebuttal letter, since the line specifications are probably not correct.

Abstract:

Perhaps it is due to the limitation of the words that may be used, but it is not immediately apparent to the reader that both the T2-weighted images and the contrast-enhanced MRI images were compared with FET-PET. This is a great pity, and in this respect should be further addressed. In addition, the aim of the study should be sharpened in the abstract.

"Here we prospectively collected over 300 serial biopsy specimens from 23 patients with contrast-enhancing adult-type diffuse gliomas using a hybrid PET-MRI scanner to compare both imaging modalities" is very general.

Introduction:

As a PET/MR user, I am aware of the advantage of hybrid imaging. However, the authors should point out that a subsequent fusion of the MRI with the FET-PET images from conventional scanners is very well feasible and so in this case the additional benefit of PET/MR hybrid imaging is mainly in patient comfort.

Results:

The wording in the results section is too general.

Figure 1: An understandable Figure Legend is missing, so that the reader can understand which images (axial, sagittal, coronar) are displayed. The PET images are not well controlled, the arrows are not well visible, and the overall image is too small.

Figure 3 The denomination of the groups in the Sankey Diagram is not understandable. Figure Legend 3 should be revised. The designation of the Targets should match with the wording within the manuscript.

Table 1 is not clear to me.

How can the authors measure an SUV value in an MR study?

How were the SUV values and the TBR measured. I think the explanations in the text are not precise enough and only descriptive.

How was the size of the VOI or ROI defined, according to what criteria.

Which SUV value was determined SUVmax or Mean; The same applies to TBR!

Methodology:

In my opinion, the data acquisition and rationale of this is not adequately described.

I agree with the authors that early FET-PET might be sufficient, however I wonder how a full diagnostic MRI with T2, T1 and a T1 with contrast agent will be acquired in 10 min acquisition time. The authors report late imaging (after 60 min), but the results of this are not presented, or the rationale to do late imaging.

The evaluation of PET as well as MR images is not adequately described.

A qualitative decision for the choice of the biopsy site does not seem adequate to me. Why was not the area with the highest FET uptake chosen? At least I cannot understand the criteria. This should be adapted accordingly in the manuscript.

Here no descriptions about SUV calculations (mean or max) are given. Also for the calculation of TB first, no methods are pointed out. A qualitative determination for the definition of the biopsy site seems to me insufficient for a prospective study. Rather, the authors should specify the TBR (SUVmax/background). Here it would be advantageous to at least indicate the value in the hotspot.

For each patient the area with the most aggressive finding in histology should be mentioned and the semiquantitative data shown.

RESPONSE TO REVIEWERS' COMMENTS

Reviewer #3, with expertise in glioma, imaging

A reviewed manuscript was available for evaluation which investigated the value of FET-PET-MR in 23 patients with a heterogenous group of patients with brain tumors.

Thank you for highlighting this point, truly, after biopsy the group occurred not to be homogenous (including lower- and higher-grade gliomas) which is not very surprising. However, patients disqualified from surgery were qualified before biopsy based on criteria suggesting high grade gliomas and all with contrast enhancement and other radiological characteristics of diffuse adult type gliomas.

Here, early FET images were compared with corresponding contrast-enhanced and T2-weighted images and correlated with tissue sampling from stereotactic biopsy.

Overall, this is an interesting study that provides some additional evidence to what is already known from the available literature. Even though the manuscript seems interesting in principle, in my opinion major revisions seem necessary:

First, I must point out that in the revised manuscript, the changes have unfortunately not been marked, as is normally the case. In this respect, I can unfortunately also not understand the corrections mentioned in the rebuttal letter, since the line specifications are probably not correct.

Please find all changes marked in red in the new version of manuscript.

Abstract:

Perhaps it is due to the limitation of the words that may be used, but it is not immediately apparent to the reader that both the T2-weighted images and the contrast-enhanced MRI images were compared with FET-PET. This is a great pity, and in this respect should be further addressed. In addition, the aim of the study should be sharpened in the abstract.

“Here we prospectively collected over 300 serial biopsy specimens from 23 patients with contrast-enhancing adult-type diffuse gliomas using a hybrid PET-MRI scanner to compare both imaging modalities” is very general.

We make the above-mentioned sentence more informative and pointed out that both the T2-weighted images and the contrast-enhanced MRI images were compared with FET-PET.

Introduction:

As a PET/MR user, I am aware of the advantage of hybrid imaging. However, the authors should point out that a subsequent fusion of the MRI with the FET-PET images from conventional scanners is very well feasible and so in this case the additional benefit of PET/MR hybrid imaging is mainly in patient comfort.

Thank you for this comment. I have added this additional advantage to the introduction.

Results:

The wording in the results section is too general.

Figure 1: An understandable Figure Legend is missing, so that the reader can understand which images (axial, sagittal, coronar) are displayed. The PET images are not well controlled, the arrows are not well visible, and the overall image is too small.

We have modified the figure to improve overall size of the images. We have added arrows to improve visibility of targeted areas and added the axis definition, SUV, TBR values and HP results to the image.

Figure 3 The denomination of the groups in the Sankey Diagram is not understandable. Figure Legend 3 should be revised. The designation of the Targets should match with the wording within the manuscript.

We have modified Figure 3 to improve clarity of the Sankey plot. The denomination of the groups is now identical to manuscript. Figure legend is also modified and clearer.

Table 1 is not clear to me.

We have added an explanation to the figure. Table 1. Basic characteristics of the study participants. Patient are divided in columns according to tumor grade. Differences between each tumor grade in terms of age, sex, previous treatment, overall survival, SUV and TBR values. Patients were similar in terms to basic characteristics in each tumor grade with exemption of age (patients diagnosed with G4 were older, $p=0.015$). SUV and TBR values were similar in T1-GAD and PET and significantly lower in FLAIR.

How can the authors measure an SUV value in an MR study?

The SUV value was measured in PET images. In iPlan software (BrainLab) we used dedicated application for PET/MRI full volumetric image fusion to verify SUV in every target point in relation to MRI.

How were the SUV values and the TBR measured. I think the explanations in the text are not precise enough and only descriptive.

How was the size of the VOI or ROI defined, according to what criteria.

A spherical VOI with a diameter of 30 mm in the contralateral hemisphere including white and gray matter, as published previously [Lohmann,Unterrainer] was used in the study.

Lohmann P, Herzog H, Rota Kops E, Stoffels G, Judov N, Filss C, et al. Dual-time-point O-(2-[18F]fluoroethyl)-L-tyrosine PET for grading of cerebral gliomas. *Eur Radiol.* 2015;25(10):3017–24

Unterrainer M, Vettermann F, Brendel M, Holzgreve A, Lifschitz M, Zähringer M, Suchorska B, Wenter V, Illigens B, Bartenstein P, Albert N. Towards standardization of 18F-FET PET imaging: Do we need a consistent method of background activity assessment? *EJNMMI Research.* 2017, 7. 10.1186/s13550-017-0295-y.

Which SUV value was determined SUVmax or Mean; The same applies to TBR!

We used SUVmax for each targeted area and the mean physiological brain activity uptake in healthy appearing cortex of the hemisphere contralateral to the tumour including grey and white matter. TBRmean was calculated as ration of SUVmax/VOI in contrallateral brain .

Methodology:

In my opinion, the data acquisition and rationale of this is not adequately described.

I agree with the authors that early FET-PET might be sufficient, however I wonder how a full diagnostic MRI with T2 , T1 and a T1 with contrast agent will be acquired in 10 min acquisition time.

It is a very important point. The MRI aquisition was done after PET early acquisition. All MRI sequences are done before late PET acquisition 40-60 min a.r.i.

The authors report late imaging (after 60 min), but the results of this are not presented, or the rationale to do late imaging.

Indeed, we have done late imaging but for this particular study only early images were evaluated and used for biopsy planning. Standard procedure in our Department of Nuclear Medicine is to perform early and late images but to improve study quality we used only early images as those were not standard worldwide and the value of early acquisition was still a matter of debate in time of planning the study. Before planning the protocol for this study we have published a paper in *radither oncol* in and the reviewer raised the argument that early FET image may not represent an active tumor and we have found it interesting to verify in prospective biopsy study that we present here.

The evaluation of PET as well as MR images is not adequately described.

All PET and MRI examinations were performed simultaneously on a 3-tesla PET/MR Siemens Biograph mMR (syngo MR E11 ver.). A standard MR imaging protocol comprising T2 TSE (T2WI, transverse, repetition time/echo time = 6000/96), T2 BLADE COR (T2WI, coronal, repetition time/echo time = 5500/118), T2 TIRM DARK FLUID (fluid-attenuated inversion recovery-FLAIR, repetition time/echo time = 6000/94), T2 SWI 3D (susceptibility weighted imaging, 3-dimensional, repetition time/echo time = 2200/2.5), DIFFUSION (diffusion-weighted imaging, B0, B1000, ADC, , repetition time/echo time = 7100/95), T1 TSE TRA (T1WI, transverse, repetition time/echo time = 500/9), T1 TSE GAD (T1WI, transverse, sagittal, coronal, repetition time/echo time = 500/9) + CM, T1 MPRAGE 3D GAD (T1WI (magnetization Prepared rapid gradient echo), 3-dimensional, repetition time/echo time = 2200/2,48) + CM was used.

¹⁸F-FET was produced and applied as described previously [Czy mamy metodę w jakimś artykule opisaną]. All patients fasted for at least 4 h before the examination as recommended, and approximately 200 MBq of ¹⁸F-FET was injected intravenously. FET-PET acquisition started 5-10 min after tracer injection and 55-60 min, acquisition time 600s, algorithm HD-PET (OSEM 3D-PSF, iterative-point spread function, 3 ITERATIONS, 21 SUBSETS). Relative scatter method and filter gaussian (FWHM 2.0mm) were used.

A qualitative decision for the choice of the biopsy site does not seem adequate to me. Why was not the area with the highest FET uptake chosen? At least I cannot understand the criteria. This should be adapted accordingly in the manuscript.

In a first visual analysis, qualitative evaluation was performed and the lesion of interest was classified as either positive, when tracer uptake visually exceeds the background activity in the contralateral cortex. The biopsy target points were selected by analyzing the uptake value in relation to contrast enhancement and the uptake in surrounding areas. We have not selected an exact threshold (not known up to date in terms of early acquisition) rather than increase uptake in relation to surrounding brain. If the hotspot was found outside the contrast enhancement the area have been selected.

In our previous study we have found a major differences in T1-Gad and early FET-Pet volumes therefore we prespecified this target before the study.

Here no descriptions about SUV calculations (mean or max) are given. Also for the calculation of TBR first, no methods are pointed out. A qualitative determination for the definition of the biopsy site seems to me insufficient for a prospective study. Rather, the authors should specify the TBR (SUV_{max}/background). Here it would be advantageous to at least indicate the value in the hotspot. For each patient the area with the most aggressive finding in histology should be mentioned and the semiquantitative data shown.

Thank you for this comment. We have described the SUV and TBR calculation. Also, TBR's at the hot spot are now presented in the table for each patient in relation to histology.

REVIEWERS' COMMENTS:

Reviewer #3 (Remarks to the Author):

I have read the answers in the Rebuttal Letter.

Unfortunately, not everything as noted in the rebuttal letter has been incorporated into the manuscript (SUV in Figure 1??), moreover, some points could well have been incorporated into the discussion.

I am also not convinced methodologically, since, for example, the references given for the FET-PET evaluation (Unterrainer et al.) are not followed.

I also maintain that the explanation of Table 1 is still not clear. If you read it like this, you think that SUV was measured in MR. Unfortunately, this does not correspond to the quality of the journals.

RESPONSE TO REVIEWERS' COMMENTS

1. Unfortunately, not everything as noted in the rebuttal letter has been incorporated into the manuscript (SUV in Figure 1??)

All the points noted in the previous rebuttal have now been incorporated into the manuscript. Sorry for the confusion.

All the figures have also been reformatted, reorganized, and improved to enhance the presentation of the results. We have moved the figure presenting the exact SUV point values from the Supplementary to the main manuscript. The SUV is presented in a targeted area as defined using biopsy planning software (new Figure 3).

2. Moreover, some points could well have been incorporated into the discussion.

We have now added a further comprehensive discussion regarding the advantages of hybrid PET-MRI and TBR assessment corresponding to biopsy planning:

“Hybrid PET-MRI has several advantages. First, integrated PET-MRI images are directly and accurately fused. Moreover, the use of gadolinium contrast does not interfere with PET acquisition and allows comparison of PET in relation to MRI without a time interval that may negatively impact the results. This is of particular importance in neuro-navigation and biopsy planning. Second, the procedure favors patient comfort, which is particularly important in this population with decreased neurological function. The dual time-point ¹⁸F-FET-PET protocol is of sufficient length to simultaneously acquire a multi-parametric MRI of the brain without increasing the time required to conduct standard dual time-point PET or MRI. Finally, more accurate diagnosis and optimized treatment may help to reduce costs.”

“Preliminary quantitative image analysis found that a PET-MRI intensity threshold 1.4-times greater than symmetrical brain uptake could identify “FET hot-spot” areas, and this result now needs further validation. Although “standard” images (after 40-60 min) were acquired, these results are not presented, as early images were the novel aspect of the protocol, and the value of early acquisition remained a matter of debate when planning the study. Additionally, after biopsy, the group was found not to be homogenous (including few lower-grade gliomas) due to the intrinsic heterogeneity of contrast-enhancing gliomas. However, all patients qualified for the study prior to biopsy based on criteria suggesting high-grade glioma and all with contrast enhancement and other radiological characteristics of diffuse adult-type gliomas.”

I am also not convinced methodologically, since, for example, the references given for the FET-PET evaluation (Unterrainer et al.) are not followed.

Sorry about this. We have now completely rewritten the methodology regarding PET-MRI evaluation.

Lohmann used a spherical VOI, and we followed this methodology. This approach was also presented and tested by Unterrainer (both references are now included in the revised manuscript). For further clarity of the methodological approach, we present a new Figure 3 with a background VOI image used for TBR calculation. We have also added the following discussion:

“Although background assessment is essential when evaluating TBR values and quantifying biological tumor volume, the TBR calculation is not crucial in terms of our findings. As the targeted points are usually localized in the same part of the brain and the same background value is used for all targets

(Figure 3a), differences between targets with respect to absolute SUV and TBR are similar (see Figure 1c).”

We have also explained in more detail and using a scheme how a full diagnostic MRI with T2, T1, and T1 with contrast were acquired (see timeline in Figure 3c). The PET methodology has also been updated as follows:

Pre-biopsy hybrid PET-MRI acquisition and evaluation

Patients meeting the inclusion criteria were referred for hybrid ¹⁸F-FET-PET-MRI examination in one procedure with simultaneous data collection at the Department of Nuclear Medicine, Franciszek Lukaszczyk Oncology Center, Bydgoszcz, Poland. PET was performed 5-15 minutes after administration of ¹⁸F-FET and again after about 60 minutes (dual time-point examination) as per standard procedures of the Department of Nuclear Medicine. For the ¹⁸F-FET-PET, all patients fasted for at least 4 h before PET-MRI as recommended, and a static emission recording in 3-dimensional mode was started on intravenous injection of 300 MBq of ¹⁸F-FET. The FET-PET acquisition time was 600 s, and the HD-PET (OSEM 3D-PSF, iterative-point spread function, 3 iterations, 21 subsets), relative scatter method, and filter Gaussian (FWHM 2.0 mm) algorithms were used.

Static PET data were reconstructed according to our clinical protocol using a 3-dimensional ordered-subset expectation-maximization algorithm with corrections for attenuation, scatter, random events, and dead time.

All ¹⁸F-FET-PET-MRI examinations were performed simultaneously on a 3-tesla PET/MR (Siemens Biograph mMR syngo MR E11; Siemens Healthineers) for the clinical indication. Patients underwent MRI scans with a slice thickness of 1 mm. In all patients, axial T2-weighted sequences as well as T1-weighted sequences before and after intravenous administration of a 0.1 mmol/kg bolus of gadobutrol (Gadovist [Bayer Healthcare]; injection rate of 3 mL/s, ACIST EMPOWER MR) were acquired. A dedicated ultrashort-echo-time sequence provided by the vendor was used for PET attenuation correction. The MRI acquisition was performed after early PET acquisition, and most sequences were completed before late PET acquisition at 40-60 min. A whole PET-MRI procedure showing all sequences is presented in Figure 3b,c.

A standard MR imaging protocol comprising T2 TSE (T2WI, transverse, repetition time/echo time = 6000/96), T2 BLADE COR (T2WI, coronal, repetition time/echo time = 5500/118), T2 TIRM DARK FLUID (fluid-attenuated inversion recovery-FLAIR, repetition time/echo time = 6000/94), T2 SWI 3D (susceptibility-weighted imaging, 3-dimensional, repetition time/echo time = 2200/2.5), DIFFUSION (diffusion-weighted imaging, B0, B1000, ADC, repetition time/echo time = 7100/95), T1 TSE TRA (T1WI, transverse, repetition time/echo time = 500/9), T1 TSE GAD (T1WI, transverse, sagittal, coronal, repetition time/echo time = 500/9) + CM, T1 MPRAGE 3D GAD (T1WI (magnetization prepared rapid gradient echo), 3-dimensional, repetition time/echo time = 2200/2,48) + CM was used.

¹⁸F-FET-PET image analysis included the evaluation of both static images. The PET-MR analysis conducted prior to biopsy was fully quantitative using dedicated software (Syngovia VB60, Siemens Healthcare, Erlangen, Germany). Based on VOI analysis, mean uptake values in the symmetrical brain and SUV values in voxels corresponding to target points were defined. A spherical VOI (diameter 30 mm) in the contralateral hemisphere including white and gray matter was used^{20,38}; however, the SUV in the final target was defined as a point value (see Figure 3a).

I also maintain that the explanation of Table 1 is still not clear. If you read it like this, you think that SUV was measured in MR.

We have changed Table 1. We have moved the SUV and TBR data to new **Figure 1c**. We have also changed the descriptions to clarify that SUV and TBR were measured at exact Target points defined by PET.

REVIEWERS' COMMENTS

Reviewer #3 (Remarks to the Author):

The questions have now been adequately answered by the authors. I noticed a few linguistic errors that should be checked by the editors. Also, the quality / size of the images is still very small; also here should be acted accordingly.

Some minor points:

Please correct the sentence starting in line 151

Across the entire stud,. 83 samples were taken from FLAIR areas, of which 11 were

Please correct this statement on line 187

As the targeted points are usually localized in the same part of the brain and the same background value is used for all targets (Figure 3a), differences between targets with respect to absolute SUV and TBR are similar (see Figure 1c).

This statement is only true for intraindividual comparison of one patient at one time point, but this is not true, when you compare your finding interindividually or sequentially (e.g. for a follow-up exam)

RESPONSE TO REVIEWERS' COMMENTS

The questions have now been adequately answered by the authors. I noticed a few linguistic errors that should be checked by the editors.

Thank you. We also look forward to the editors' input.

Also, the quality / size of the images is still very small; also here should be acted accordingly.- Are you able to improve it accordingly?

We have revisited the figures and ensured that they meet the journal's requirements.

Some minor points:

Please correct the sentence starting in line 151

Across the entire stud,. 83 samples were taken from FLAIR areas, of which 11 were

Thanks, this typo (and others) has been corrected.

Please correct this statement on line 187

As the targeted points are usually localized in the same part of the brain and the same background value is used for all targets (Figure 3a), differences between targets with respect to absolute SUV and TBR are similar (see Figure 1c).

This statement is only true for intraindividual comparison of one patient at one time point, but this is not true, when you compare your finding interindividually or sequentially (e.g. for a follow-up exam).

Thanks, we agree, and we have adjusted this sentence to reflect that it only applies in limited circumstances.